# Physics-Based Aircraft Dynamics Identification Using Genetic Algorithms

Raymundo Peña-García [1,†], Rodolfo Daniel Velázquez-Sánchez [1,†], Cristian Gómez-Daza-Argumedo [2,†], Jonathan Omega Escobedo-Alva [2,†], Ricardo Tapia-Herrera [3,†] and Jesús Alberto Meda-Campaña [1,*,†]

[1] Instituto Politécnico Nacional SEPI ESIME Zacatenco, Ciudad de México 07738, Mexico; rpenag1600@alumno.ipn.mx (R.P.-G.); rvelazquezs1901@alumno.ipn.mx (R.D.V.-S.)
[2] Instituto Politécnico Nacional SEPI ESIME Ticomán, Ciudad de México 07340, Mexico; cgomezdazaa1500@alumno.ipn.mx (C.G.-D.-A.); jescobedoa@ipn.mx (J.O.E.-A.)
[3] CONAHCYT—Instituto Politécnico Nacional SEPI ESIME Zacatenco, Ciudad de México 07738, Mexico; rtapiah@ipn.mx
[*] Correspondence: jmedac@ipn.mx
[†] These authors contributed equally to this work.

**Abstract:** This research introduces a physics-based identification technique utilizing genetic algorithms. The primary objective is to derive a parametric matrix, denoted as *A*, describing the time-invariant linear model governing the longitudinal dynamics of an aircraft. This is achieved by proposing a fitness function based on the properties of the transition matrix and taking advantage of some of the capabilities of the genetic algorithm, mainly those of restricting the search ranges of the unknowns. In this case, such unknowns are related to the type of aircraft and flight conditions that are considered during the identification process. The proposed identification method is validated with a reliable nonlinear model that can be found in the literature, as well as with the calculation of the trim condition and linearization generally used in aircraft dynamics. In summary, this study suggests that the genetic algorithm provided with the adequate fitness function could be an appealing alternative for aircraft model identification, even when limited data are available. Furthermore, in some cases, linearization using a genetic algorithm can be more efficient than classical methods.

**Keywords:** physical-based modeling; system identification; transition matrix; genetic algorithm

## 1. Introduction

### 1.1. Motivation

One of the fundamental pursuits in the scientific domain is the development of mathematical models that accurately represent physical systems. A fundamental criterion for evaluating such models is their usefulness. This may involve the ability to predict crucial aspects of the behavior of a physical system or to provide critical information to the design process through model parameters. In all cases, the mathematical model must achieve a balance between simplicity to be of practical use, and complexity to describe the most important characteristics of the system being modeled [1,2].

A long-standing need in aeronautics has been that of obtaining reliable and accurate models for aircraft dynamics. This goes beyond aircraft design and flight simulation; it involves the implementation and validation of control systems. In the current era, because of the impressive development of unmanned aerial vehicles (UAVs), the search for control systems has become more demanding, both in civil and military applications. The arrival of high-reliability, low-cost sensors and sophisticated control systems has broken down previous barriers. Small aircraft can now reach higher altitudes, higher speeds, and longer ranges. This provides the opportunity to implement cutting-edge algorithms, which are both smarter and more efficient [3].

However, the effort to identify an optimal model for advanced control system design objectives remains a costly and complex task. Given this reality, it is imperative to reevaluate the identification process, seeking an innovative approach that is simple, fast, and effective. It is crucial to avoid heuristic models and costly experiments, recognizing that the simplest option is often the most prudent. Several compelling and direct examples can be found in [4]. Over the past two decades, identification in aeronautics, particularly for manned airplanes, has been vigorously explored by prominent civil military and government entities such as the Deutsches Zentrum für Luft-und Raumfahrt (DLR) and NASA [5].

A correct identification system incorporates experimental data and undergoes validation in flight. Nowadays, a huge amount of information on experimental identification and validation processes can be obtained from various technical reports and manuals [6], affording insights into many facets of authentic identification procedures. Recent research [5] offers an excellent overview of the state of the art in this field. Tools for the time domain are comprehensively examined in [7], with additional contributions from the German Aerospace Center featured in [8]. Furthermore, system identification techniques for aircraft dynamics in the frequency domain have been provided in [9].

In [5], it is noted that advances in machine learning have yet to yield substantial benefits for aircraft system identification, primarily due to the absence of a physics-based model. This limitation becomes evident when employing neural networks to represent dynamics [10] or aircraft dynamics as a black box [11], or when resorting to genetic algorithms in pursuit of a heuristic model [12]. There are also published works where two techniques are combined; for instance, in [13], genetic algorithms are part of a hybrid identification system originally based on optimization, and, in [14], the blend of neural networks and fuzzy systems is considered to obtain the longitudinal model of an aircraft.

When the integration of artificial intelligence is judiciously combined with an understanding of physics-based aircraft models, along with the aforementioned state-of-the-art knowledge, it enables the imposition of parametric constraints that facilitate the use of swift convergence algorithms. This approach circumvents the conventional issue of infinite solutions in computational algorithms and yields more cost-effective results compared to relying solely on artificial intelligence or physics-based algorithms [15]. As demonstrated, this robust identification methodology proves invaluable not only for linear dynamics identification but also for the linearization under diverse trim conditions. It offers a simple, effective, and tailored solution to the formidable challenges encountered in today's aerospace industry.

It is generally recognized that the identification of nonlinear models poses a greater challenge compared to linear models. For example, successful identification through neural networks requires an appropriate determination of the number of layers, neurons per layer, and learning factor, all depending on the characteristics of the nonlinear system under consideration. Similarly, for fuzzy systems, an optimal approximation depends on the selection of an appropriate number of fuzzy rules, together with a judicious choice of membership functions. Even in the case of SINDY, customization is often imperative to address the unique complexities associated with the nonlinear system being identified [16]. On the other hand, when the mentioned techniques are applied to identify linear systems, the problem reduces to parametric identification [17].

While a large number of approaches are available in the literature to successfully identify both linear and nonlinear systems, this paper focuses specifically on the utilization of a practical and effective technique based on genetic algorithms to linearize the longitudinal dynamics of an aircraft by introducing a new fitness function.

### 1.2. Contribution

A notable disadvantage of the application of genetic algorithms in obtaining mathematical models from real or simulated data is that, during the evaluation of the fitting function, the model obtained must be simulated in each iteration to determine the error between the data and the response of the intermediate model. This causes a very high com-

putational cost that may be out of the reach of researchers with limited budgets. To reduce this computational cost, and considering that the linear system is sufficient to describe the longitudinal dynamics of an aircraft, in this work, the transition matrix [18,19] is considered instead of the complete simulation of the intermediate model in each iteration of the genetic algorithm.

Therefore, the main goal of the current work is that of presenting a fitness function based on the transition matrix, such that, when the fitness function is included in the genetic algorithm, the matrix, *A*, involved in the linear approximation of the aircraft, is promptly obtained, and relatively few data are needed.

*1.3. Manuscript Organization*

The rest of this paper is organized as follows. In Section 2, the nonlinear model of the aircraft is presented. Section 3 describes the trim condition considered. In Section 4, the traditional linearization of the nonlinear model is performed. Below, Section 5 presents a concise analysis of the parameters included in matrix *A*. The main result, i.e., the linearization achieved by the genetic algorithm, is discussed in detail in Section 6. Finally, in Section 7, some conclusions are presented.

## 2. Nonlinear 6-DOF Dynamic Model of a Research Aircraft

The primary function of a nonlinear model is to represent and capture complex relationships between variables that do not follow a linear pattern, allowing greater flexibility in modeling intricate patterns and dependencies. It is important to mention that, although in the current work only the longitudinal dynamics are excited, it is necessary to pay attention to the complete dynamics of the aircraft. Next, the nonlinear model used in the analysis and validation of the identification process of the genetic algorithm is presented synthetically. This complete and validated model was extracted from [20], which is an excellent summary of [21].

In accordance with rigid body dynamics and classical approximation from Newton–Euler, the following analysis can be carried out:

Translation equation on body axes:

$$\dot{V}^b_{cm/e} = \frac{1}{m} F^b - \omega^b_{b/e} \times V^b, \tag{1}$$

where $m$ is the mass, the sub-index $cm/e$ means from the center of mass to Earth axis, and super-index $b$ indicates body frame; this nomenclature comes from kinematics equations deduction (see [22]), with $\dot{V}^b_{cm/e} = [\dot{u}\ \dot{v}\ \dot{w}]^T$ as the linear acceleration vector, $V^b_{cm/e} = [u\ v\ w]^T$ as the linear velocity vector, $F^b$ as the external forces, $\dot{\omega}^b_{b/e} = [\dot{p}\ \dot{q}\ \dot{r}]^T$ as the angular acceleration, and $\omega^b_{b/e} = [p\ q\ r]^T$ as the angular rate.

Rotational equations on body axes:

$$\dot{\omega}^b_{b/e} = (J^b)^{-1} \left[ M^b - \omega^b_{b/e} \times J^b \omega^b_{b/e} \right] \tag{2}$$

where $J^b$ is the inertial matrix, and $M^b$ are the external moments.

Kinematic Euler Equation:

At this point, it is important to consider a transformation of angular rates from body axes to the general frame Euler system (inertial), as follows:

$$\dot{\Phi} = H(\Phi)\dot{\omega}^b_{b/e}, \tag{3}$$

where $\dot{\Phi} = [\dot{\phi}\ \dot{\theta}\ \dot{\psi}]^T$ is the vector of Euler angles rate and

$$H(\Phi) = \begin{bmatrix} 1 & sen(\phi)tan(\theta) & cos(\phi)tan(\theta) \\ 0 & cos(\phi) & -sen(\phi) \\ 0 & sec(\theta)sen(\phi) & cos(\phi)sec(\theta) \end{bmatrix}.$$

The full nonlinear model is considered nonautonomous and depends on the full state (even forces and moments are direct functions of the states) and control inputs, resulting in the following:

$$\dot{X} = f(X, U), \tag{4}$$

with the state vector

$$X = \begin{bmatrix} x_1 & \dots & x_{12} \end{bmatrix}^T = \begin{bmatrix} u\ v\ w\ p\ q\ r\ \phi\ \theta\ \psi\ P_N\ P_E\ \bar{h} \end{bmatrix}^T,$$

where $P_N$, $P_E$, and $P_D$ are the $x - y - z$ aircraft displacements on North–East–Down navigation frame system and $\bar{h} = -P_D$.

As mentioned above, external forces and moments are functions of states and control inputs, resulting in the following:

$$F^b(X, U),$$

$$M^b(X, U),$$

which ultimately leads to

$$\begin{bmatrix} \dot{x}_1 \\ \dot{x}_2 \\ \dot{x}_3 \\ \\ \dot{x}_4 \\ \dot{x}_5 \\ \dot{x}_6 \\ \\ \dot{x}_7 \\ \dot{x}_8 \\ \dot{x}_9 \\ \\ \dot{x}_{10} \\ \dot{x}_{11} \\ \dot{x}_{12} \end{bmatrix} = \begin{bmatrix} \frac{1}{m} F^b(X, U) - \begin{bmatrix} x_4 \\ x_5 \\ x_6 \end{bmatrix} \times \begin{bmatrix} x_1 \\ x_2 \\ x_3 \end{bmatrix} \\ \\ (J^b)^{-1} \left[ M^b(X, U) - \begin{bmatrix} x_4 \\ x_5 \\ x_6 \end{bmatrix} \times J^b \begin{bmatrix} x_4 \\ x_5 \\ x_6 \end{bmatrix} \right] \\ \\ H(\Phi) \begin{bmatrix} x_4 \\ x_5 \\ x_6 \end{bmatrix} \\ \\ CS(\phi, \theta, \psi) \begin{bmatrix} u \\ v \\ w \end{bmatrix} \end{bmatrix}. \tag{5}$$

In Equation (5), $CS(\phi, \theta, \psi)$ is the body to navigation axis transformation:

$$\begin{bmatrix} c\theta c\psi & (-c\phi s\psi + s\phi s\theta c\psi) & (s\phi s\psi + c\phi s\theta c\psi) \\ \\ c\theta s\psi & (c\phi c\psi + s\phi s\theta s\psi) & (-s\phi c\psi + c\phi s\theta s\psi) \\ \\ -s\theta & s\phi c\theta & c\phi c\theta \end{bmatrix}, \tag{6}$$

where $c(\cdot)$ and $s(\cdot)$ are *cosine* and *sine* function, respectively.

*The Aircraft*

The original airplane is a twin-engine research civil aircraft with a weight mass of 120,000 kg and inertial moment $I_y$ of $7.68 \times 10^6$ kg $\cdot$ m$^2$, a mean aerodynamic chord of 6.6 m, and a wing platform area of 260 m$^2$. The dynamics of the control input and the allowable limits on the elevator are specified as $-15 \le \dot{\delta}_e \le 15$ degrees per second and $-25 \le \delta_e \le 10$ degrees, respectively. For the two motors, the rate and end limits are $-1.6 \le \dot{\delta}_{th} \le 1.6$ degrees per second and $0.5 \le \delta_{th} \le 10$ degrees.

## 3. Trim Condition

In the context of identification, the equilibrium condition will be a steady-state flight with wings level and no sideslip [22]. Determining the equilibrium point in a coupled, non-

linear, multivariable system, such as Equation (9), is nontrivial and requires a specialized algorithm [23]. Before linearization, a fitting condition for the system must be selected in the general form $\dot{X} = f(X, U)$:

$$f(X_e, U_e) = 0. \tag{7}$$

In accordance with Equation (5), navigation variables $X_e \in \mathbb{R}^9$ and $U_e \in \mathbb{R}^5$ are neglected, i.e., $\dot{u} = \dot{v} = \dot{w} = \dot{p} = \dot{q} = \dot{r} = 0$ for steady-state flight. At this point, the methodology presented in [22] is considered, whose problem statement is the following:

$$\min_{X \in \Omega} \hat{f}(X) = f(X) + C^T H C, \tag{8}$$

where $H$ is a weight diagonal matrix, and $C \in \mathbb{R}^{9+1}$ is the restriction vector. The process is then performed with the Nelder and Mead Simplex minimization search algorithm, using the command *fminsearch()* MATLAB, which is a nonlinear programming solver to find the minimum of an unconstrained multivariable function using the derivative-free method. The cost function and initial lookup values are required. In this case, all functions in $C$ that are equal to zero are

$$C = \begin{bmatrix} \dot{x}_1 & \dots & \dot{x}_9 & (|V^b_{cm/e}| - V_e) & \gamma & x_2 & x_7 & x_9 \end{bmatrix}; \tag{9}$$

in Equation (9), $V_e$ is the required operation speed in trim condition, $|V^b_{cm/e}| = \sqrt{x_1^2 + x_2^2 + x_3^2}$, and $\gamma = x_8 - atan(x_3/x_1)$ is the flight path angle. The weight matrix was defined as $H = diag(3)$ after a trial and error process that provided valid results. The final function cost is $\hat{f}(X) = C^T H C$.

Using *fminsearch()* with the cost function defined above, after 9 iterations and with a minimization error of $1 \times 10^{-19}$, the clipping condition $X_e$ obtained is $x_1 = 110$ m/s, $x_3 = -2.0882$ m/s, $x_8 = -0.0634$ rad, and all remaining states are zero.

## 4. Linearization

For the coefficient identification process, it is necessary to obtain the time-invariant linear system around the equilibrium point $X_e$. To this end, one may consider Equation (4) with $\dot{x} \in \mathbb{R}^n, x \in \mathbb{R}^n, u \in \mathbb{R}^m$, and the mapping function $f : \mathbb{R}^{n+m} \longrightarrow \mathbb{R}^n$. This expression can be rewritten in an implicit form as follows:

$$F(\dot{x}, x, u) = \dot{x} - f(x, u) = 0, \tag{10}$$

where $f : \mathbb{R}^{2n+m} \longrightarrow 0^n$. By linearizing Equation (10) through a Taylor expansion series around the equilibrium point and considering only low-order terms (linear), the following results are obtained:

$$F(\dot{x}, x, u) = F(\dot{x}_0, x_0, u_0) + \frac{\partial F}{\partial \dot{x}}(\dot{x}_0, x_0, u_0)(\dot{x} - \dot{x}_0)$$

$$+ \frac{\partial F}{\partial x}(\dot{x}_0, x_0, u_0)(x - x_0) + \frac{\partial F}{\partial u}(\dot{x}_0, x_0, u_0)(u - u_0). \tag{11}$$

Now, the equilibrium condition,

$$F(\dot{x}_0, x_0, u_0) = 0, \tag{12}$$

must be evaluated. Then, from Equations (10) and (12), $\dot{x} = \dot{x}_0, x = x_0, u = u_0$, and define the partial derivatives as

$$\frac{\partial F}{\partial \dot{x}}(\dot{x}_0, x_0, u_0) = E, \tag{13}$$

$$\frac{\partial F}{\partial x}(\dot{x}_0, x_0, u_0) = \hat{A}, \tag{14}$$

$$\frac{\partial F}{\partial u}(\dot{x}_0, x_0, u_0) = \hat{B}, \tag{15}$$

and, by simplifying Equation (11), one has

$$0 = E\delta\dot{x} + \hat{A}\delta x + \hat{B}\delta u. \tag{16}$$

Thus, solving for $\delta\dot{x}$ yields the following:

$$\delta\dot{x} = A\delta x + B\delta u, \tag{17}$$

where

$$A = -\frac{\hat{A}}{E}, \tag{18}$$

$$B = -\frac{\hat{B}}{E}. \tag{19}$$

It is important to mention that the LTI system $\dot{x} = Ax + Bu$ with matrices Equations (18) and (19) will provide values of perturbations ($\delta x$) from an initial equilibrium condition $X_e$, as in the case of the control inputs.

The partial derivatives appearing in Equations (13)–(15) can be obtained by using a numerical algorithm that employs a finite differential approximation, a feasible approach in simulation where one has control over the reliability and integrability of the data. While the process is relatively straightforward, careful consideration must be afforded to numerical errors, which can be mitigated by avoiding excessively small values of $h$. In the literature, it is widely acknowledged that central differentiation is more accurate compared to one-sided difference formulas [24,25]. Then, for any partial derivatives, one has

$$\frac{\partial F(t_k)}{\partial x(t_k)} = \frac{F(x(t_{k+1})) - F(x(t_{k-1}))}{2h}, \tag{20}$$

where $h = x(t_{k+1}) - x(t_k)$. This method is considered to determine $\hat{E}$, $\hat{A}$, and $\hat{B}$ column by column, i.e.,

$$\begin{aligned} E_{i,j} &= \frac{F_i(\dot{x}_j(t_{k+1})) - F_i(\dot{x}_j(t_{k-1}))}{2h_j}, \\ \hat{A}_{i,j} &= \frac{F_i(x_j(t_{k+1})) - F_i(x_j(t_{k-1}))}{2h_j}, \\ \hat{B}_{i,j} &= \frac{F_i(u_j(t_{k+1})) - F_i(u_j(t_{k-1}))}{2h_j}. \end{aligned} \tag{21}$$

To obtain data and calculate the matrices in Equation (21), the initial condition and the input signals are used as perturbations. For instance, to produce a response in all states of the nonlinear model with no input signal, $x(0)_1 = 10$ m/s and $x(0)_2 = 5$ m/s are chosen as initial conditions; these values are additional to the trim condition. Then, after an increment of the state variable $x_j$ from $h_j = 1 \times 10^{-12}$, the function $F_i$ is computed from each state $x_j$ to finally calculate the partial derivatives that make up matrices as in Equation (21). According to Equations (18) and (19), the state-space matrices for the 6-degree-of-freedom linear system are determined. Such a system is described by the differential equation $\dot{x} = Ax + Bu$, where $x \in \mathbb{R}^9$ and $u \in \mathbb{R}^5$. However, the longitudinal dynamics can be separated as in the following reduced system:

$$A = \begin{bmatrix} -0.052 & -0.0011 & 6.852 & -9.7903 \\ -0.2333 & -0.9104 & 107.9602 & 0.6215 \\ -0.0044 & -0.0431 & -1.4537 & 0 \\ 0 & 0 & 1 & 0 \end{bmatrix}, \tag{22}$$

$$B = \begin{bmatrix} 0 & -0.8022 & 0 & 9.8100 & 9.8100 \\ 0 & -12.6387 & 0 & 0 & 0 \\ 0 & -5.0424 & 0 & 0.3924 & 0.3924 \\ 0 & 0 & 0 & 0 & 0 \end{bmatrix}. \tag{23}$$

The matrix $A$ is directly related to Equation (24) and is considered parametric, implying a physical interpretation.

## 5. Parameters

The matrix coefficient in a state-space form has a well-known analytic expression in the field of flight dynamics. In this particular context, only longitudinal dynamics are considered. The matrix $A$ is square and corresponds to the vector state $x = [u\ w\ q\ \theta]^T$. Each coefficient is assigned a specific nomenclature, delineating moments and forces as functions of the state variables, i.e.,

$$A = \begin{bmatrix} X_u & X_w & X_q & X_\theta \\ Z_u & Z_w & Z_q & Z_\theta \\ M_u & M_w & M_q & 0 \\ 0 & 0 & 1 & 0 \end{bmatrix}. \tag{24}$$

In Equation (24), $X_\theta$ and $Z_\theta$ represent the linearized force projections due to gravity on body axes and are direct functions of the trim condition defined by $\theta_e$; $M$ refers to moments on body axes, specifically controlling pitch. Data for the common coefficient range parameters and their physical interpretation can be found in various sources, including [22,26–28]. These significant values are contingent upon the specific aircraft and prevailing flight conditions.

## 6. Main Result: Linearization by Means of the Genetic Algorithm (GA)

In this section, the nonlinear model introduced in Section 2 is employed as the benchmark for the evaluation of the performance of the linearization methods outlined in Section 4, including the genetic algorithm-based linearization approach. The maneuver considered corresponds to a cruise flight condition wherein the aircraft is traveling at a speed of 122.8357 m/s at sea level. The specified state variables include an angle of attack of 1.1354 degrees, a pitch angle of −3.7701 degrees, and all remaining states set to zero. The trim elevator input is −5.9505 degrees, and both motors operate at the 67.5% of total power. Utilizing the nonlinear model defined by Equations (1)–(6), the longitudinal behavior of the aircraft is simulated, deviating from the trim condition with perturbations of $\Delta u_0^{nl} = 10$ m/s, $\Delta \alpha_0^{nl} = 2.3309$ degrees, $\Delta q_0^{nl} = 12$ degrees per second, and $\Delta \theta_0^{nl} = 0$ degrees over a duration of $t = 180$ s with a sampling time of $T = 0.05$ s.

In order to establish a reference for comparing the genetic algorithm (GA) linearization, the linear system with matrix $A$, derived through the linearization procedure detailed in Section 4 and represented by Equation (22), is simulated under identical conditions. The numerical results are illustrated in Figure 1, considering only perturbation data from the trimmed flight condition above. The mean squared error (MSE) between the linear states and their corresponding nonlinear counterparts is computed as 0.0034.

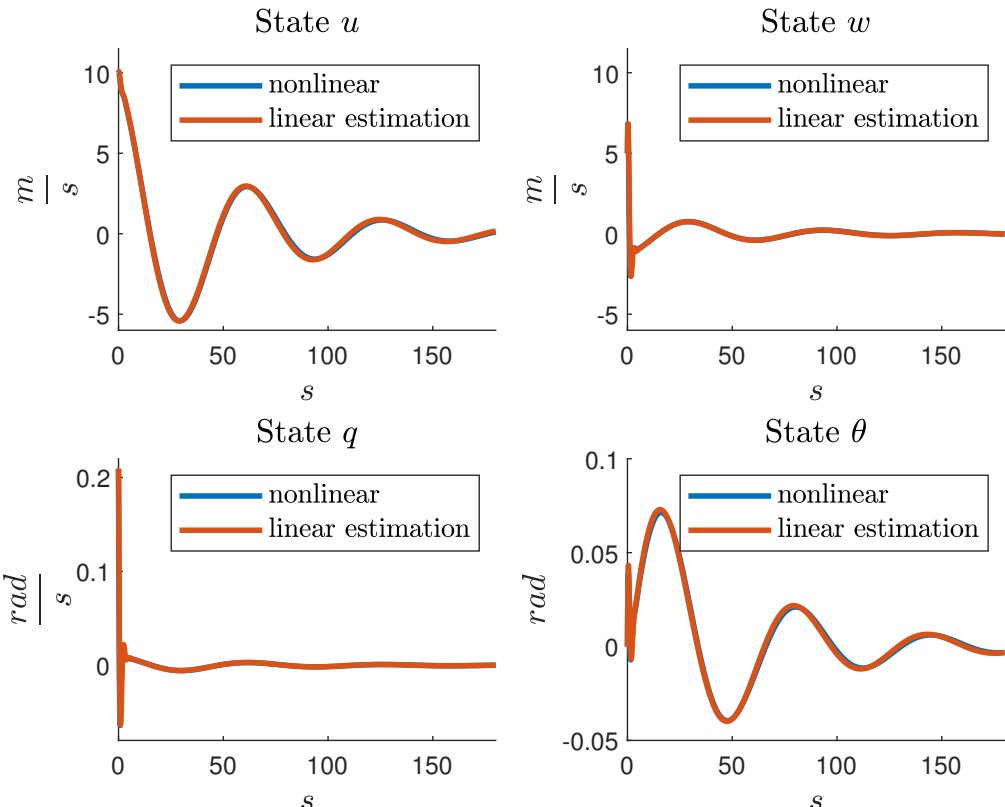

**Figure 1.** Simulated data with an initial condition located away from the equilibrium point. The linearization process, as described in Section 4, is applied to obtain the corresponding linearized representation.

Next, the nonlinear system is linearized using a genetic algorithm. Before presenting the results, a concise description of the genetic algorithm is provided.

### 6.1. Genetic Algorithm (GA)

A genetic algorithm (GA) is an optimization technique that mimics the processes of evolution and natural selection. Usually, it is applied to solve complex optimization problems.

In a genetic algorithm, a potential solution to the problem is represented as a chromosome, typically as a binary string, but other representations are also possible, depending on the problem at hand. A population is a collection of such chromosomes. The size of the population is a user-defined parameter. The fitness function evaluates how well each chromosome in the population solves the problem. It assigns a fitness score to each chromosome based on the goal of the problem. *The fitness function is problem-specific and provided by the user.*

Chromosomes with higher fitness scores have a greater chance of being selected for reproduction. The most common selection methods include roulette wheel selection and tournament selection. Crossover is the process of combining two parent chromosomes to create one or more offspring. Various crossover operators are used, such as one-point, two-point, and uniform crossover. The crossover rate (probability of crossover) is a user-defined parameter. Mutation involves randomly changing some bits in a chromosome with a low probability. It introduces genetic diversity into the population. The mutation rate (probability of mutation) is another user-defined parameter. Genetic algorithms terminate when a stopping criterion is met. Common termination conditions include a maximum number of generations, reaching a satisfactory fitness level, or a time limit.

A typical genetic algorithm operates in a loop, which includes selection, crossover, mutation, and replacement of the old population with the new one. This process continues

for a predefined number of generations [29–33]. It is important to mention that the GA has been successfully used to solve a number of problems in aeronautics. For instance, a methodology involving a multiobjective genetic algorithm for exploring parametrized microstructures is presented in [34]. In [35], the authors investigate the aerodynamics of desert locust tandem wings through computational fluid dynamics (CFD) simulations. Using 2D and 3D Navier–Stokes equations, they explore wing interactions and corrugation effects. Optimization with genetic algorithms and Nash game theory improves gliding performance by at least 77%. In [36], an automated framework for developing interpolatable aeroelastic reduced-order models (AE ROMs) across diverse flight conditions is presented. By combining system identification, state-consistency enforcement (SCE), and a genetic algorithm (GA), the approach addresses the issue of state inconsistency in ROMs.

In MATLAB, optimization via genetic algorithm (GA) can be performed using the function ga, which is part of the Global Optimization Toolbox. The syntax is as follows:

$$x = ga(fun, nvars, A, b, Aeq, beq, lb, ub, nlcon, opts), \tag{25}$$

where $fun$ is the fitness function to be optimized; $x$ is the vector of variables included in the fitness function; $nvars$ is the dimension of the vector $x$; the pair $(A, b)$ defines the inequality restrictions in the form of $A \cdot x \leq b$; the pair $(Aeq, beq)$ defines the equality restrictions in the form of $Aeq \cdot x = beq$; the vectors $lb$ and $ub$, both of dimension $nvars$, define the lower and upper limits for the search of $x$, respectively; $nlcon$ defines the nonlinear constraints; $opts$ allows the user to modify parameters such as tolerance, the type of plot to be depicted, the maximal number of generations, etc. For more details, please refer to [37].

An essential challenge when working with GA is the need to define an appropriate fitness function. This is a crucial factor as the fitness function plays a critical role in evaluating how effectively a solution addresses the specific problem. The selection of the fitness function can significantly influence the overall performance of the algorithm and the quality of the solutions it produces.

With this in mind, a simple yet practical fitness function is introduced in the following sections. The function is designed to derive a parametric matrix $A$ that characterizes the longitudinal dynamics of an aircraft. It utilizes data obtained from the simulation of the highly reliable nonlinear aircraft model described in Section 2.

*6.2. Linearization via GA*

In this section, the objective is to find a matrix $A$ in the form of Equation (24) for the linearization of the longitudinal dynamics of the aircraft described in Equations (1)–(6) using genetic algorithms (GA). To accomplish this, a fitness function is derived in a practical manner. The unknown $x's$ values to be determined by the GA are organized in matrix $A_{GA}$ as follows:

$$A_{GA} = \begin{bmatrix} x(1) & x(4) & x(7) & -9.7903 \\ x(2) & x(5) & 107.9602 & x(9) \\ x(3) & x(6) & x(8) & 0 \\ 0 & 0 & 1 & 0 \end{bmatrix}, \tag{26}$$

where the constants appearing in Equation (26) are fixed values dependent on the characteristics of the modeled aircraft. In other words, only nine of the sixteen elements of the matrix regarding Equation (26) need to be determined.

It is proposed that the fitness function evaluates the matrix described in Equation (26) in each iteration, comparing the linear states with their corresponding nonlinear states at arbitrary sample instants. To minimize computational load, the inclusion of the transition matrix is considered in this process.

6.2.1. Transition Matrix

The transition matrix, denoted as $\Phi(t - t_0) = e^{A(t-t_0)}$, describes the evolution of a time-invariant linear system over time, considering a given initial condition $x(t_0)$. It shows

how the state of the system at an initial time $t_0$ is transformed to the state at a later instant $t$. For continuous-time systems described by $\dot{x}(t) = Ax(t)$, the transition matrix can be used to obtain $x(t)$ from the initial condition $x(t_0)$ (with $t_0 = 0$) as follows:

$$x(t) = \Phi(t,0)x(0). \tag{27}$$

Please refer to [18,19] for further details. In MATLAB, the transition matrix can be computed using the function $expm()$, and its application in this work is explained below.

### 6.2.2. Fitness Function

For optimized computational resource utilization, the proposal is to apply the fitness function to a reduced set of samples. Rather than utilizing the entire set of simulated data, consisting of 3601 data points collected over a duration of 180 s with a sampling interval of $T = 0.05$ s, a smaller subset must be considered. Such a subset is suggested to comprise the region where the linear approximation is desired to closely match the nonlinear data.

Taking this into account, from the simulation of the nonlinear model Equations (1)–(6) and based on the transition matrix mentioned above, the following algorithm describes the fitness function used to linearize the nonlinear dynamics of the aircraft.

Explanation:

- Line 1: The name of the function is $fitness$, and the argument $x \in \mathbb{R}^9$ is the vector of the variables to be found by the GA in order to minimize this function.
- Line 2: Global variables $ndata$, $x_{NL}$, $x_0$, and $t$ are used in this function, where $ndata$ is the number of samples considered during the linearization, $x_{NL} \in \mathbb{R}^{ndata \times 4}$ is a matrix containing the values of the nonlinear states $u, w, q, \theta$, from the numerical simulation at desired instants, $x_0$ represents the vector of initial conditions, and $t \in \mathbb{R}^{ndata}$ is a vector with the corresponding time instants at which matrix $x_{NL}$ has been obtained.
- Line 3: Matrix $A_{GA}$ is constructed from elements of vector $x$ from a previous iteration.
- Line 4: Matrix $x_L \in \mathbb{R}^{ndata \times 4}$ is initialized as a matrix of zeros with $ndata$ rows (number of samples) and four columns (states $u, w, q, \theta$).
- Lines 5–7: A loop iterates over the elements of $t$, using the exponential matrix operation to compute the states of $x_L$ at each one of the time instants included in vector $t$.
- Lines 8–11: Squared errors $sqe_i$ for each state, with $i = 1, \ldots, 4$.
- Line 12: The total error $f$ is computed as the square root of the sum of squared errors.
- Line 13: The function returns $f$ as the result of the fitness evaluation.

### 6.3. Numerical Simulations

In this section, the results of two linearizations using the genetic algorithm (GA) are presented. These experiments were performed on a personal computer with the following specifications: Intel(R) Core(TM) i5-10500H CPU @ 2.50 GHz, 16 GB of RAM, running MATLAB 2019b.

### 6.3.1. Simulation 1: GA Linearization Using 31 Samples

In this case, only the samples obtained during the first 3 s of the nonlinear simulation with a sampling interval of $T = 0.1$ s are considered. This involves selecting every other sample from the nonlinear simulation, resulting in a total of 31 samples. In this example, according to the fitness function outlined in Algorithm 1, one has $ndata = 31$, $x_{NL} \in \mathbb{R}^{31 \times 4}$, $x_0 = [1050.20940]^T$, and $t \in \mathbb{R}^{31}$.

---

**Algorithm 1** Function *fitness*

---

1: **procedure** FITNESS($x$)
2:     **global** $ndata, x_{NL}, x_0, t$
3:     $A_{GA} \leftarrow \begin{bmatrix} x(1) & x(4) & x(7) & -9.7903 \\ x(2) & x(5) & 107.9602 & x(9) \\ x(3) & x(6) & x(8) & 0 \\ 0 & 0 & 1 & 0 \end{bmatrix}$
4:     $x_L \leftarrow \text{zeros}(\text{size}(t), 4)$
5:     **for** $k = 1$ **to** $ndata$ **do**
6:         $x_L(k, :) \leftarrow \text{expm}(A_{GA} \cdot t(k)) \cdot x_0$
7:     **end for**
8:     $sqe_1 \leftarrow (x_{NL}(1) - x_L(1))^T \cdot (x_{NL}(1) - x_L(1))$
9:     $sqe_2 \leftarrow (x_{NL}(2) - x_L(2))^T \cdot (x_{NL}(2) - x_L(2))$
10:    $sqe_3 \leftarrow (x_{NL}(3) - x_L(3))^T \cdot (x_{NL}(3) - x_L(3))$
11:    $sqe_4 \leftarrow (x_{NL}(4) - x_L(4))^T \cdot (x_{NL}(4) - x_L(4))$
12:    $f \leftarrow \sqrt{sqe_1 + \ldots + sqe_4}$
13:    **return** $f$
14: **end procedure**

---

During the execution of the GA in MATLAB, neither inequality nor equality restrictions nor nonlinear constraints are considered. However, due to the limitations of the available computer resources, the decision was made to search for the solution around the values obtained by the linearization procedure explained in Section 4.

Therefore, the GA in MATLAB was executed using the following three lines of code:

$$\begin{aligned} opts &= optimoptions('ga',' PlotFcn', @gaplotbestf) \\ fun &= @fitness \\ x &= ga(fun, 9, [\,], [\,], [\,], [\,], lb, ub, [\,], opts) \end{aligned} \tag{28}$$

with

$$\begin{aligned} lb &= [-0.06, -0.3, -0.005, -0.002, -1, -0.05, 5.5, -2, 0] \\ ub &= [0, 0, 0, 0, 0, 0, 7, 0, 0.7], \end{aligned} \tag{29}$$

It is worth mentioning that the options chosen in Equation (28) are primarily intended for monitoring the GA's behavior during execution, which may lead to a higher consumption of computer resources. A more refined selection of these parameters could potentially yield more efficient results. For a comprehensive description of MATLAB's function *ga*, interested readers are referred to [37].

The elapsed time is 355.863055 s, and the matrix obtained by the GA is

$$A_{GA1} = \begin{bmatrix} -0.0572 & -0.0020 & 5.5 & -9.7903 \\ -0.2367 & -0.9198 & 107.9602 & 0.0001 \\ -0.005 & -0.05 & -1.5393 & 0 \\ 0 & 0 & 1 & 0 \end{bmatrix}. \tag{30}$$

Then, matrix Equation (30) is used to simulate the linear system for 180 s with a sampling time of $T = 0.05$ s in order to compare the states of the linear identified system with the nonlinear ones. The results are presented in Figure 2. The overall mean square error (MSE) between the nonlinear states and their linear approximations is 0.0167.

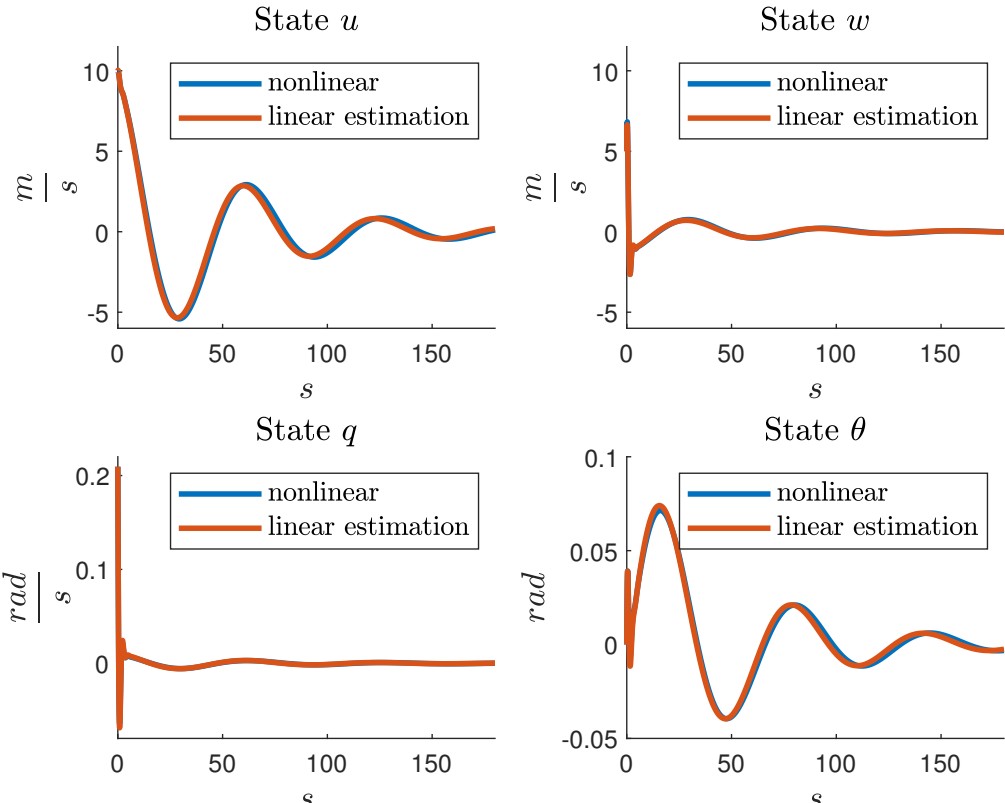

**Figure 2.** Simulated data with initial condition located away from the equilibrium point. Linearization achieved via GA using the first three seconds of nonlinear simulated data, sampled at $T = 0.1$ s.

Notice that the MSE obtained in this experiment is greater than the one obtained by the linearization of Section 4, but the GA has been executed considering only 31 samples. In the next experiment, a larger set of samples is considered.

6.3.2. Simulation 2: GA Linearization Using 66 Samples

Now, the samples that occurred during the initial 3 s of the nonlinear simulation with a sampling time of $T = 0.1$ s, along with samples taken from the remainder of the signal with $T = 5$ s, are considered, resulting in a total of 66 samples. Therefore, one has $ndata = 66$, $x_{NL} \in \mathbb{R}^{66 \times 4}$, $x_0 = [10 \, 5 \, 0.2 \, 0.940]^T$, and $t \in \mathbb{R}^{66}$. The execution of the GA in MATLAB is exactly the same as presented in Equations (28) and (29). The elapsed time is 1304.814475 s, and the matrix obtained by the GA is

$$A_{GA2} = \begin{bmatrix} -0.0549 & -0.0019 & 6.1047 & -9.7903 \\ -0.2209 & -0.9005 & 107.9602 & 0.0001 \\ -0.0046 & -0.0499 & -1.5690 & 0 \\ 0 & 0 & 1 & 0 \end{bmatrix}. \tag{31}$$

As before, the linear system with state matrix Equation (31) is simulated for 180 s with a sampling time of $T = 0.05$ s, and the comparisons with the nonlinear states are depicted in Figure 3. The overall mean square error (MSE) between the nonlinear states and their linear approximations is notably smaller: $4.3984 \times 10^{-4}$, but the execution time is larger than in the previous case.

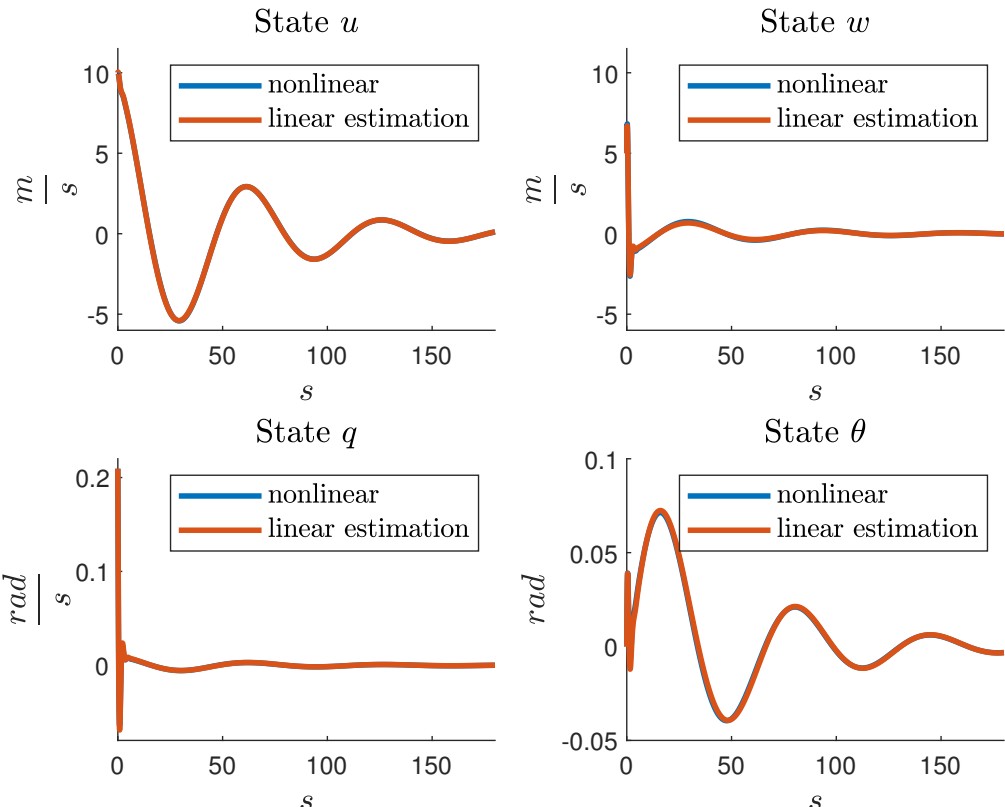

**Figure 3.** Simulated data with initial condition located away from the equilibrium point. Linearization achieved using a GA using the initial three seconds of nonlinear simulated data, sampled at $T = 0.1$ s, followed by $T = 5$ s for the rest of the nonlinear data.

### 6.4. Linearization via Least Squares (LS) Method

In contrasting the potential of the genetic algorithm (GA), a presentation of a simple linear regression utilizing the least squares method is provided. It is widely recognized that perfect identification results when applying this method to data generated by a linear time-invariant (LTI) system without noise or external perturbations. However, complications in identification may arise when applying this method to data from a nonlinear system [7]. The method was, nonetheless, applied using the same nonlinear simulation data employed in the GA algorithm. Initially, the analysis was conducted using a limited dataset comprising 31 samples, with a final time of 3 s and samples taken at 0.1 s intervals. The matrix obtained is presented in Equation (32), and the results of the numerical simulation are depicted in Figure 4. The overall MSE between the nonlinear states and their linear approximations is $6.2725 \times 10^6$.

$$A_{LQ1} = \begin{bmatrix} 0.003 & -0.103 & 10.4008 & 14.6247 \\ -0.0805 & -1.0208 & 118.5216 & 1.5095 \\ -0.0032 & -0.0489 & -1.5903 & 0.2304 \\ 0 & 0 & 1 & 0 \end{bmatrix}. \tag{32}$$

Finally, the entire dataset produced by the nonlinear simulation was considered, spanning 180 s with a sample interval of 0.05 s. In this case, the matrix obtained is shown in

Equation (33), while the numerical results are illustrated in Figure 5. In this case, the overall MSE is 235.2556.

$$A_{LQ_2} = \begin{bmatrix} -0.001 & -0.0095 & 2.3994 & 0.21 \\ -0.0056 & -0.037 & 3.3085 & 0.2203 \\ 0 & -0.004 & 0.0231 & 0.0026 \\ 0 & 0.0001 & 0.0018 & 0.0049 \end{bmatrix}. \tag{33}$$

This analysis is summarized in Table 1.

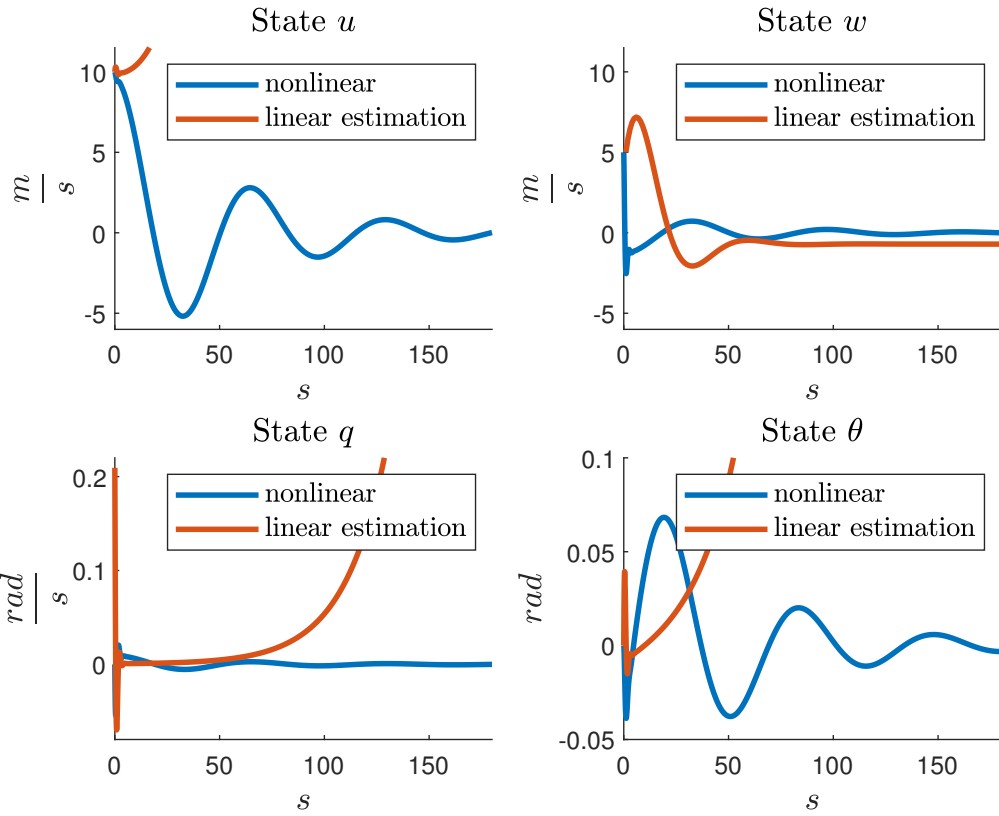

**Figure 4.** Linearization achieved using linear regression with least squares method using the initial three seconds of nonlinear simulated data, sampled at $T = 0.1$ s.

**Table 1.** MSE and execution time.

| Linearization via | MSE | Execution Time |
|:---:|:---:|:---:|
| Section 4 | 0.0034 | —— |
| GA (31 samples) | 0.0167 | 355.86 s |
| GA (66 samples) | $4.3984 \times 10^{-4}$ | 1304.81 s |
| LS (31 samples) | $6.2725 \times 10^{6}$ | —— |
| LS (full dataset) | 235.2556 | —— |

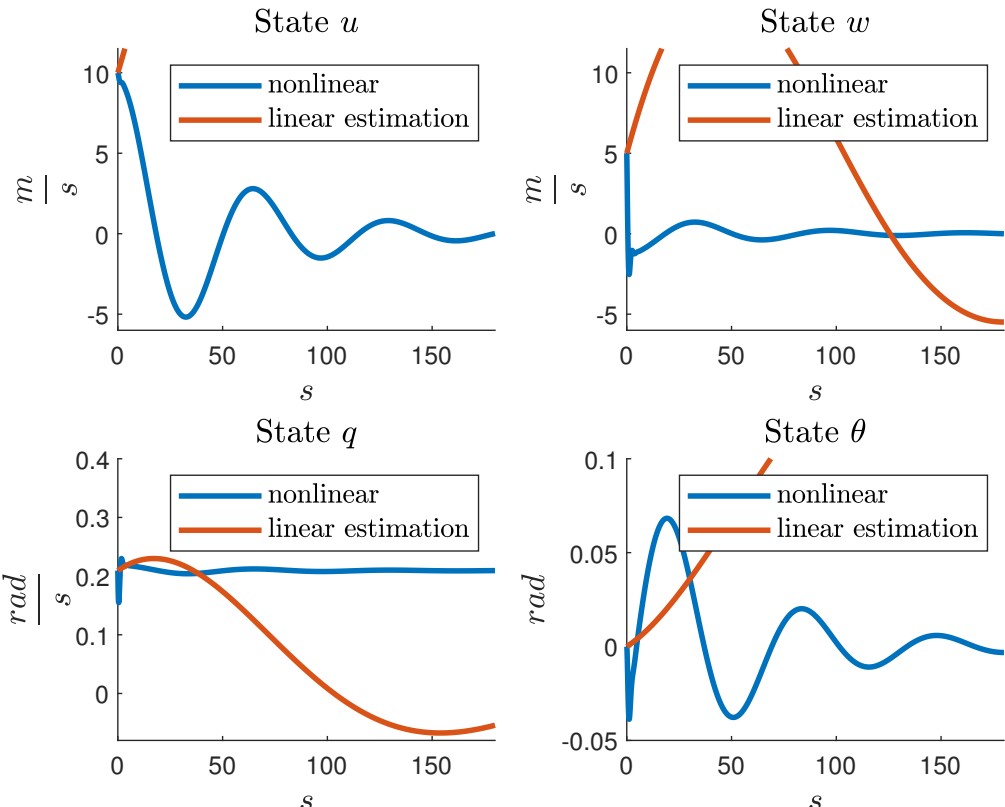

**Figure 5.** Linearization accomplished through linear regression employing the least squares method, utilizing the complete dataset of the nonlinear simulated data sampled at $T = 0.05$ s.

## 7. Conclusions

The traditional linearization algorithm is based on the idealized assumption of perfectly clean and complete data, often considering $h = 1 \times 10^{-12}$, rendering it impractical for real-world experimental data. Nonetheless, it has been demonstrated in this study that the genetic algorithm (GA) not only generates matrices with clear physical interpretations but also exhibits the essential robustness necessary for reliable experimental identification. Furthermore, the automation of the linearization process is achievable.

In situations characterized by limited computational resources, where applying GA from scratch to linearize nonlinear dynamics is impeded by unknown search ranges, the GA approach based on the proposed transition matrices proves valuable as a post-processing tool. This is particularly advantageous when measurements of the nonlinear system are available, as demonstrated in this study, showcasing its capacity to enhance results obtained through traditional linearization methods.

Most identification methodologies, including the one presented here, can demonstrate robustness against noise when appropriately designed. With this proposal, assurance is provided that matrix $A$ possesses a physical interpretation, and the presence of nonlinearity does not impede the linear identification process.

In contrast to other methodologies that face challenges in deriving a matrix $A$ with a clear physical interpretation, this approach excels in this regard. Furthermore, the model obtained through this methodology proves functional across a broader operational zone compared to alternative methods.

Considering these aspects, this approach emerges as an alternative for identifying linear time-invariant (LTI) equations from experimental data. The exploration of this potential, particularly in broader experimental scenarios, constitutes an essential avenue for future research.

**Author Contributions:** Conceptualization, J.O.E.-A. and J.A.M.-C.; formal analysis, R.P.-G., R.D.V.-S., C.G.-D.-A., J.O.E.-A. and R.T.-H.; investigation, J.O.E.-A., R.T.-H. and J.A.M.-C.; methodology, J.O.E.-A., R.T.-H. and J.A.M.-C.; project administration, J.A.M.-C.; writing—original draft, J.O.E.-A. and J.A.M.-C.; writing—review and editing, J.A.M.-C. All authors have read and agreed to the published version of the manuscript.

**Funding:** This research received no external funding.

**Data Availability Statement:** Data are contained within the article.

**Acknowledgments:** This work was supported in part by Consejo Nacional de Humanidades, Ciencias y Tecnologías (CONAHCYT), through the Scholarship Sistema Nacional de Investigadores (SNI); in part by Instituto Politécnico Nacional through Research Project under Grant 20230023 and 20231126; in part by the Scholarship Estímulo al Desempeño de los Investigadores (EDI); in part by the Scholarship Comisión de Operación y Fomento de Actividades Académicas (COFAA); and in part by the Scholarship Beca de Estímulo Institucional de Formación de Investigadores (BEIFI).

**Conflicts of Interest:** The authors declare no conflicts of interest.

## Notation and Definitions

| | |
|---|---|
| $A$ | Matrix for LTI longitudinal dynamics |
| $\hat{A}$ | Jacobian matrix of nonlinear equations with respect to the state variables |
| $B$ | Input Matrix for LTI system |
| $\hat{B}$ | Jacobian matrix of nonlinear equations with respect to inputs |
| $c(\cdot)$ | Cosine function |
| $C$ | Matrix for output definition in state form |
| $E$ | Jacobian matrix of nonlinear equations with respect to the first-order state variables |
| $F^b$ | Force vector in body frame |
| $\hat{F}$ | Jacobian matrix of nonlinear equations with respect to the first-order state variables |
| $h$ | Step size for numerical derivative, $h = x(t_{k+1}) - x(t_k)$ |
| $\bar{h}$ | Aircraft flying height |
| $H$ | Weight diagonal matrix for optimization criteria |
| $H(\Phi)$ | Transformation matrix of angular rates from body axes to the inertial system |
| $J^b$ | Inertial tensor of body center of gravity |
| $m$ | Aircraft mass |
| $M^b$ | Moment vector in body axes |
| $M_q$ | Moment due to pitch rate |
| $M_u$ | Moment due to horizontal velocity in body axes |
| $M_w$ | Moment due to vertical speed in body axis |
| $p$ | Roll rate in x body axis |
| $\dot{p}$ | Angular acceleration in the $x$-axis in body frame |
| $P_E$ | Aircraft displacement to the east direction (navigation frame) |
| $P_N$ | Aircraft displacement to north direction (navigation frame) |
| $q$ | Pitch rate in y body axis |
| $\dot{q}$ | Angular acceleration in the $y$-axis in body frame |
| $r$ | Yaw rate in body axes |
| $\dot{r}$ | Angular acceleration on the $z$-axis in body frame |
| $s(\cdot)$ | Sine function |
| $T$ | Step size for genetic algorithm |
| $u$ | Flight velocity in x-axes in body frame from the equilibrium state (cruise flight) |
| $U$ | Input vector control |
| $\dot{u}$ | Flight acceleration in x-axes in the body frame and from the equilibrium state (cruise flight) |
| $U_e$ | Input vector control for equilibrium condition |
| $u_0$ | Input for equilibrium condition in the linear system |
| $v$ | Lateral component for flight speed from equilibrium condition, in body frame |
| $V_{cm/e}^b$ | Aircraft velocity from the center of mass to Earth axis, super-index $b$ indicates body frame |
| $V_e$ | Flight velocity in trim condition |
| $\dot{v}$ | Lateral component for acceleration from equilibrium point, in body frame |

| | |
|---|---|
| $w$ | Vertical component for flight velocity from equilibrium condition, in body frame |
| $\dot{w}$ | Vertical component for acceleration from equilibrium condition, in body frame |
| $X$ | Twelve dimensional state vector |
| $x$ | x displacement on body frame |
| $x_0^{nl}$ | Initial condition for non linear model |
| $X_e$ | Vector state in trim condition |
| $X_\theta$ | Forces in x body axis due to pitch angle |
| $x_0$ | State values in equilibrium point for the linear system definition |
| $\dot{x}_0$ | First-order state values in equilibrium point for the linear system definition |
| $\dot{X}$ | First-order vector state |
| $x_{NL}$ | Matrix containing the values of the nonlinear states |
| $X_q$ | Forces in x body axis due to pitch rate |
| $X_u$ | Forces in the x-body axis due to velocity in the x-axis of the body frame |
| $X_w$ | Forces in the x-body axis due to velocity in the z-axis of the body frame |
| $y$ | Lateral displacement in body axis |
| $z$ | Vertical displacement in body axis |
| $Z_\theta$ | Forces in z-body axis due to pitch angle |
| $Z_q$ | Forces in z-body axis due to pitch rate |
| $Z_u$ | Forces in the z-body axis due to velocity in the x-axis of the body frame |
| $Z_w$ | Forces in the z-body axis due to velocity in the z-axis of the body frame |
| $\gamma$ | Longitudinal flight path angle |
| $\theta$ | Pitch angle |
| $\theta_e$ | Pitch angle for trim condition |
| $\dot{\theta}$ | Pitch rate |
| $\Phi$ | Vector of Euler angle rates, angular rates of aircraft respect to the inertial system |
| $\phi$ | Roll angle |
| $\dot{\phi}$ | Roll rate |
| $\Phi(t - t_0)$ | Transition matrix $\Phi(t - t_0) = e^{A(t-t_0)}$ |
| $\Phi(t, 0)$ | Transition matrix with initial condition $x(0) = 0$ |
| $\psi$ | Yaw angle |
| $\dot{\psi}$ | Yaw rate |
| $\omega_{b/e}^b$ | Angular rate vector in body axis |
| $\dot{\omega}_{b/e}^b$ | Angular acceleration vector in body axis |

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
