# Peer review of "Physics-Based Aircraft Dynamics Identification Using Genetic Algorithms"

_aerospace, doi:10.3390/aerospace11020142_

Round 1

Reviewer 1 Report

Comments and Suggestions for Authors

Dear Author,

Thank you for submitting your paper. While reviewing your work, I believe there is significant potential in your research. However, I have some suggestions for modifications to further enhance the quality and professionalism of your paper.

Comparative Analysis: Your paper could benefit from a more detailed comparison between the improved model and the baseline model. Consider adding comparison charts to clearly demonstrate the superiority of the enhanced model in key metrics. Additionally, statistical methods to validate the significance of the improvements should be employed and thoroughly explained in the manuscript.

Conclusion Section: The conclusion section appears overly concise. In this section, you could summarize your research findings, emphasize the superiority of the enhanced model over the baseline, and further discuss the potential impact of these improvements on practical applications. Ensure that your conclusion has sufficient depth and insight for readers to fully grasp the significance of your research.

Comments on the Quality of English Language

Language Refinement: Throughout the paper, ensure the use of professional and accurate terminology, avoiding overly colloquial expressions. Check for and correct any grammar and spelling errors. Each sentence should be clear and unambiguous to the reader.

Reviewer 2 Report

Comments and Suggestions for Authors

The authors demonstrated the identification of aircraft longitudinal state-space matrices using a genetic algorithm (GA). Although the topic is not original, the problem is of interest to the field, because the use of genetic algorithm techniques might be interesting in some cases, particularly in nonlinear system identification.

The computational time for execution seems to be somewhat large, and the boundaries for each parameter supplied to the algorithm were rather narrow, requiring the use of previous knowledge about the model parameters, which limits the applicability and usefulness of this approach. 

Moreover, there are other approaches for this type of problem that the authors could have explored, such as gradient-based optimizations, in order to assess the advantages of the genetic algorithm. The authors did not perform validation tests, to assess the robustness of each model identified. This analysis would have enriched the results of the paper. In general, the discussions are somewhat shallow and must be improved with additional analyses.

More details about the content and the format are given below:

The use of state transition matrix to compute simulations is not efficient. The experimental data is already discrete, so a more straightforward solution would be to discretize the system using c2d.m and then use the discrete-time formulation of the state-space equations. Or, more directly, simply use the linear simulation function "lsim.m". This would make algorithm 1 much more efficient, without the need for skipping some of the data points (lines 5-7 in the algorithm).

In Section 6, Fig.1, the maneuver should be described. It seems that this is simply the simulation (with zero inputs) starting from a non-trimmed condition, which causes oscillations at t<5s, but this is very not clear in the paper. Depending on the type of maneuver, some parameters will be easier to identify than others.

Page 14, line 414

"ndata = 9"

ndata is the number of samples used (66), not 9. Please verify.

About the results and conclusion:

Page 15, line 425

"our study demonstrates that GA not only produces matrices with clear physical interpretations"

Note that the physical interpretations came from the formulation of the optimization problem. This would happen to any algorithm used, i.e. not a feature of the GA.

The MSE in Section 6.3.2 of 4.3984E-4 was unexpected. The GA results were based on a single maneuver, so it would be interesting to check using other maneuvers for validation of the model. I suspect that the model identified by the GA works better than the linearization for this maneuver alone, but might not work as well for other maneuvers (or different initial conditions).

Additional questions:

What are the confidence intervals of the identified parameters? Can the GA provide estimates for the standard errors?

It would be interesting to check how the optimization method using GA performs when compared to a more conventional optimization using fminsearch, for example. Is it faster? What are the advantages/disadvantages?

Comments regarding the format:

It is unusual to present "1.4. Notation and Definitions" at the end of the Introduction section. I would recommend placing it after the abstract, before the introduction.

Section 2: (page 5 line 166)

"This complete and validated model was extracted from [16], which is an excellent summary of [17]."

Even though the model is available, it is helpful to the reader to describe it briefly. What type of aircraft, what are the inputs, what are the main capabilities and limitations of the model, etc.

It should be clear in Figs 1,2,3 that the states shown are actually variations around the trim condition. For some states, the trim condition is zero (e.g. pitch rate), but that is not the general case.

Comments on the Quality of English Language

Page 1 Line 27

"The arrive"

The arrival

Page 5 Line 169

"we present the next equations"

we present the equations next 

Reviewer 3 Report

Comments and Suggestions for Authors

The paper, to my understanding, focuses on linearisation of a slightly nonlinear model. The generic linear model, mentioned in line 252, has seemingly been excited only with initial conditions and no external forces (perhaps it is why B has been neglected). Within operation of the genetic algorithm (GA), the linear model has been simulated through transition matrix (another sign for free/no-input response), and the result has been compared with the one of a complete nonlinear model to produce the value of the fitness function required in GA. By the way, these are mostly my guesses based on some key words and my familiarity with the area. It is quite difficult to read and follow the paper, and it has basic writing mistakes. The paper includes lots of barely relevant materials and lacks some critical information.

Here are some comments:

1-     Least Square of Errors (LSE), which is very much faster than GA, provides the best linear estimation of any system. I suggest that you consult with chapter 3 of Nonlinear System Identification by Oliver Nelles. The authors should mention this point, not to mislead their readers, and justify why they picked GA over LSE.

2-     The first paragraph of 1.1 needs references.

3-     The justification of the choice of a simplified linear model in the end of 1.1 is barely relevant.  Linearity or nonlinearity of the system makes no difference in parameter tuning genetic algorithm. GA can be used easily to identify nonlinear systems as reported in "An Enhanced Physics-based Model to Estimate the Displacement of Piezoelectric Actuators", Intelligent Material Systems and Structures, Vol. 26, Issue 11, Pages 1442–1451, 2015. The authors should mention this point, not to mislead readers, and justify why they decided to use GA to identify a linear model rather than a nonlinear model.

4-     The introduction is too long. Sections 1, 2 and 3 can be easily cut into half with no harm.

5-     Section 2 and initial parts of section 3 are recommended to be drastically shortened, as they mostly present existing knowledge. The nonlinear and the linear models need to be presented only, which are almost lost in many hardly relevant equations and sentences in the present form of the paper.

6-     In the title of 6.3, I suggest that Experiments is replaced with Simulations.

Comments on the Quality of English Language

1-     Pronoun “We” should not exist in a journal paper, and the actions of the authors should be presented with passive verbs.

2-     For the completed actions, which are reported in the paper, past tense should be used. Use of present tense, like in this paper, wrongly implies that, at the time of writing, the authors were planning or conducting research and had no results.

Round 2

Reviewer 1 Report

Comments and Suggestions for Authors

I think it has reached the publishing level of this journal.

Reviewer 2 Report

Comments and Suggestions for Authors

The authors have addressed the concerns raised in the previous report.

Reviewer 3 Report

Comments and Suggestions for Authors

The paper is in a better shape now; I recommend the paper for publication. 

Comments on the Quality of English Language

My previous comments have been responded satisfactorily.